# CONTRAGEN: EFFECTIVE CONTRASTIVE LEARNING FOR CAUSAL LANGUAGE MODEL

## ABSTRACT

Despite exciting progress in large-scale causal language models, the expressiveness of its representations is largely limited by the *anisotropy* issue where the hidden representations are distributed into a narrow cone in the vector space. To resolve this problem, we present CONTRAGEN, a novel contrastive learning framework at both token-level and sequence-level. We assess CONTRAGEN on a wide range of downstream tasks and show that CONTRAGEN can effectively enhance both isotropy and discrimination of the representations. This leads to the desired improvement on various language understanding tasks, which helps bridge the gap with the encoder-only models and makes causal language models more suited for tasks beyond language generation. Specifically, we attain $44\%$ relative improvement on the Semantic Textual Similarity tasks and $34\%$ on Code-to-Code Search tasks. Furthermore, by improving the expressiveness of the representations, CONTRAGEN also boosts the source code generation capability with $9\%$ relative improvement on execution accuracy on the HumanEval benchmark. [1]

## 1 INTRODUCTION

Causal Language Models (CLM) have seen remarkable success in language generation, both in natural language (Radford et al., 2018; 2019; Brown et al., 2020) and programming language (Chen et al., 2021; Nijkamp et al., 2022). However, one limitation at its core is the *anisotropy* issue, where the hidden representations, especially those output from the final layers of transformers, are squeezed into a tiny cone in the representation space. As observed by Ethayarajh (2019), the average cosine similarity of any two words from two randomly sampled sequences is almost at one when evaluating at the last hidden layer of GPT-2 (Radford et al., 2019). Such anisotropic often causes large performance gap (see Appendix D.1) with the encoder-based models on the discriminative tasks and hence limits the wide usage of CLM beyond language generation.

Two main reasons have been posited for causing the degenerated representations. Dong et al. (2021) theoretically prove that the key ingredient of language models – *self-attention* (Vaswani et al., 2017) possesses a strong inductive bias towards the representation collapse. Another line of work argues that the autoregressive next token prediction objective is the main cause for the non-isotropic representations. We thereby face a conundrum. On the one hand, both self-attention and next token prediction are the driving force behind the widespread successes of CLM on language generation (Lewis et al., 2020; Radford et al., 2019; Chen et al., 2021; Chowdhery et al., 2022). On the other hand, as shown in Section 4 and Appendix D.1, the resulting anisotropic representations cause inferior performance on language understanding tasks, *e.g.,* Semantic Textual Similarity (Agirre et al., 2012; 2013; 2014; Marelli et al., 2014; Agirre et al., 2015; 2016; Cer et al., 2017) and Code Search (Lu et al., 2021; Huang et al., 2021; Guo et al., 2022).

We tackle this challenge by leveraging contrastive learning (Chen et al., 2020; He et al., 2020; Gao et al., 2021). Intuitively, by separating each instance apart from each other, contrastive learning can produce more uniformly distributed representations. Previous studies (Wang & Isola, 2020; Wang & Liu, 2021) have shown that uniformity is important for yielding discriminative representations. We thereby present CONTRAGEN, a novel contrastive learning framework at both token-level (CONTRAGEN-TOK) and sequence-level (CONTRAGEN-SEQ). Our goals are two-fold. First, an

---

[1]We will release our code and checkpoints after the final decisions of acceptance are out.

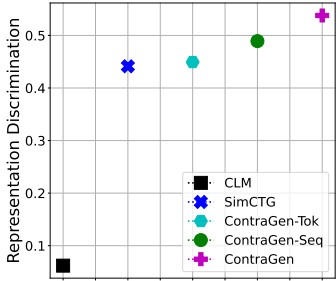

| Model | Discrimination | Generation | |
|---|---|---|---|
| | STS-AVG | Pass@1 | Rank@1 |
| CLM | 27.83 | 13.42 | 15.48 |
| SimCTG | 36.19 | 13.26 | 15.24 |
| CONTRAGEN-SEQ | 37.04 | 12.96 | 15.24 |
| CONTRAGEN-TOK | 42.03 | 13.64 | 16.99 |
| CONTRAGEN | **45.32** | **14.63** | **17.07** |

Figure 1: CONTRAGEN and its components enhance (**Left**) the representation discrimination approximated as *one minus the ratio* of *inter-similarity or uniformity* (the cosine similarity between tokens from randomly sampled sequences) and *intra-similarity* (the cosine similarity between a sequence and its constituent tokens) – high discrimination score is better; and (**Right**) the performance of discrimination (STS) and generation (HumanEval) tasks (details in Section 4).

ideal CLM should be able to better leverage the representation space by dispersing apart semantically different tokens or sequences. Second, we aim to attain separable features by mapping tokens or sequences under similar contexts to distinct but comparatively closer locations in the vector space.

We conduct extensive experiments and analyses to assess the effectiveness of our proposed framework with particular focus to address the following – (1) Does CONTRAGEN improve the *discrimination ability* of representations learned by CLM? (2) Do the enhanced representations lead to better performance on *language generation* tasks? (3) Is the joint contrastive learning at both token-level and sequence-level necessary and how do they benefit from each other? (4) How does the impact of CONTRAGEN vary across language domains?

## 2 RELATED WORK

**Anisotropic Representation of Language Models** Despite the remarkable success achieved by large scale language models (Devlin et al., 2019; Radford et al., 2019; Yang et al., 2019; Raffel et al., 2020; Lewis et al., 2020), they suffer from the *anisotropy* issue where the representations are distributed into a tiny cone in the vector space (Gao et al., 2019; Ethayarajh, 2019; Li et al., 2020; Wang et al., 2020). In particular, Ethayarajh (2019) shows that the degeneration is much severer on CLM, where the average cosine similarity between two words sampled from randomly selected sequences is almost at one when evaluating at the output from the last hidden layer of GPT-2 (Radford et al., 2019). Inspired by the previous findings (Arora et al., 2017; Mu & Viswanath, 2018), several efforts have focused on regularizing the representations to isotropic distribution either via postprocessing (Su et al., 2021; Li et al., 2020) or directly optimizing variant regularization terms during training (Gao et al., 2019; Wang et al., 2020). Although promising improvement has been observed, performance is still inadequate.

**Contrastive Learning** An alternative comes to light as the popular contrastive learning techniques Chen et al. (2020); He et al. (2020) have seen remarkable successes in Natural Language Processing (NLP). A large amount of research has focused on sentence representation learning for the encoder-only model, with the main differences lying in how the augmentations are generated (Fang & Xie, 2020; Giorgi et al., 2021; Wu et al., 2020; Meng et al., 2021; Yan et al., 2021; Kim et al., 2021; Gao et al., 2021; Zhang et al., 2022). Recently there is emerging interest in developing effective contrastive learning approach for text generation models. However, most existing work mainly focus on the encoder-decoder structure Dong et al. (2019); Raffel et al. (2020); Lewis et al. (2020) by contrasting the suboptimal model generation, obtained via diverse sampling (An et al., 2022) or adding perturbations on the embedding space (Lee et al., 2021), against the ground truth. On the other hand, it is not intuitive and even elusive to develop effective contrastive learning strategy for the decoder-only models. A recent work (Su et al., 2022) proposes, SimCTG, a token-level contrastive learning approach that aims to separate each token apart from others within the same sequence by a prefixed distance. However, we find that our temperature-based token-level contrastive learning approach, CONTRAGEN-TOK, consistently outperforms SimCTG across different tasks. We conjecture that

the fixed margin based contrastive learning allows less flexibility for the token-level representation separation within each sequence, especially considering that the semantic relevance among tokens can vary across contexts (sequences).

**Code Generation and Beyond** Language modeling for source code is an actively growing area of research. Various model architectures have been explored recently, including the encoder-only (Feng et al., 2020; Guo et al., 2021), encoder-decoder (Ahmad et al., 2021; Wang et al., 2021; Li et al., 2022), and decoder-only models (Chen et al., 2021; Nijkamp et al., 2022; Chowdhery et al., 2022). Among them, the decoder-only models found effective on the code generation front. However, as shown in Section 4.3.2 and Appendix D.1, they suffer from the less satisfied performance on the discriminative tasks (Lu et al., 2021; Huang et al., 2021; Guo et al., 2022). It motivates us to improve the decoder-only models on the discriminative tasks so as to extend its main usage beyond language generation. Furthermore, code is fundamentally different from natural language in that it is more structured, which helps validate the generalizability of our approach beyond plain text.

## 3 CONTRASTIVE LEARNING FOR LANGUAGE GENERATION

Before we present the main model, we first call out the following notations that we will use throughout this section. Let $\mathbf{x} = [\mathbf{x}_1, \mathbf{x}_2, \cdots, \mathbf{x}_{|\mathbf{x}|}]$ denote a sequence with variable length $|\mathbf{x}|$, *e.g.*, a text document or a code snippet, and $\mathbf{h} = [\mathbf{h}_1, \mathbf{h}_2, \cdots, \mathbf{h}_{|\mathbf{x}|}]$ be its representation output by the last hidden layer of the decoder. For a randomly sampled batch $\mathcal{B} = \left\{ \mathbf{x}^j \right\}_{j=1}^{N}$ with $N$ sequences, we use $\mathbf{x}_i^j$ and $\mathbf{h}_i^j$ to denote the $i^{th}$ token and its representations in the $j^{th}$ sequence, respectively. Let $\mathbf{h}^j, \mathbf{h}^{j^+}$ denote the representation pair of sequence $\mathbf{x}^j$ and $\mathbf{h}_i^j, \mathbf{h}_i^{j^+}$ correspond to the representations of the $i$-th token. Such representation pairs are referred as the positive pairs in contrastive learning, which are often obtained via data augmentation.

### 3.1 LANGUAGE GENERATION MODELING

Language generation modeling is usually formulated as sequence distribution estimation over a set of examples, $\mathcal{B} = \{\mathbf{x}^j\}_{j=1}^{N}$. For tractable estimation, common practice is to factorize the joint distribution of each sequence into the product of conditional token prediction probabilities. The language generation model is then trained via maximum likelihood estimation as $\mathcal{L}_{\text{CLM}} = -\frac{1}{N} \sum_{j=1}^{N} \sum_{i=1}^{|\mathbf{x}^j|} \log p(\mathbf{x}_i^j | \mathbf{x}_{<i}^j)$ where $\mathbf{x}_{<i}^j = [\mathbf{x}_1^j, \cdots, \mathbf{x}_{i-1}^j]$ denotes the subsequence before $\mathbf{x}_i^j$ and $|\mathbf{x}^j|$ refers to the sequence length. In this paper, we mainly focus on the decoder-only Causal Language Model (CLM), though we believe our strategy can be applied to most language models.

Despite promising progress in language generation, CLM suffers from the anisotropy issue, which limits the wide usage of it beyond language generation. Ideally, to better leverage the representation space, we aim to disperse apart the representations of tokens or sequences from different contexts while simultaneously pulling close those from the similar contexts for better discrimination. To this end, we present a novel contrastive learning framework at both token-level and sequence-level.

### 3.2 TOKEN-LEVEL CONTRASTIVE LEARNING

As aforementioned, $\mathbf{h}_i^j, \mathbf{h}_i^{j^+}$ are a pair of representations for $\mathbf{x}_i^j$, the $i$-th token in the $j$-th sequence. Let $\mathcal{I}_j = \{1, 2, \ldots, |\mathbf{x}_j|\}$ denote the indices of tokens in $\mathbf{x}_j$. Further let $\tau$ denote the temperature hyper-parameter and $\diamond$ denotes the cosine similarity, *i.e.*, $\mathbf{a} \diamond \mathbf{b} = \mathbf{a}^T \mathbf{b} / \|\mathbf{a}\|_2 \|\mathbf{b}\|_2$. Then for the token-level contrastive learning, we minimize the following,

$$\mathcal{L}_{\text{CONTRAGEN-TOK}} = \sum_{j=1}^{N} \sum_{i=1}^{|\mathbf{x}^j|} - \left( \log \frac{\exp(\mathbf{h}_i^j \diamond \mathbf{h}_i^{j^+}/\tau)}{\exp(\mathbf{h}_i^j \diamond \mathbf{h}_i^{j^+}/\tau) + \sum_{t \in \mathcal{I}_j \setminus i} \left[ \exp(\mathbf{h}_i^j \diamond \mathbf{h}_t^j/\tau) + \exp(\mathbf{h}_i^j \diamond \mathbf{h}_t^{j^+}/\tau) \right]} \right.$$

$$\left. + \log \frac{\exp(\mathbf{h}_i^{j^+} \diamond \mathbf{h}_i^j/\tau)}{\exp(\mathbf{h}_i^{j^+} \diamond \mathbf{h}_i^j/\tau) + \sum_{t \in \mathcal{I}_j \setminus i} \left[ \exp(\mathbf{h}_i^{j^+} \diamond \mathbf{h}_t^j/\tau) + \exp(\mathbf{h}_i^{j^+} \diamond \mathbf{h}_t^{j^+}/\tau) \right]} \right) .$$

The above objective tries to separate the positive representation pairs of each token at distinct locations apart from the others within the same sequence. Intuitively, pushing representations of different

tokens away from each other should indeed cause them to be more uniformly distributed. On the other hand, as suggested in Section 4, the token-level contrastive learning leads to the grouping of semantically similar sequences, which we conjecture is due to implicit clustering effect of contrastive learning (Wang & Liu, 2021; Zhang et al., 2021) on tokens and so as the sequences consist of semantically similar tokens.

### 3.3 SEQUENCE-LEVEL CONTRASTIVE LEARNING

Let $\mathcal{I}_B = \{1, 2, \ldots, N\} \cup \{1^+, 2^+, \ldots, N^+\}$ denote indices of all $2N$ sequence-level representations for batch $\mathcal{B}$. The sequence-level contrastive loss then follows as

$$\mathcal{L}_{\text{CONTRAGEN-SEQ}} = \sum_{j=1}^{N} - \left( \log \frac{\exp(\mathbf{h}^j \diamond \mathbf{h}^{j^+}/\tau)}{\exp(\mathbf{h}^j \diamond \mathbf{h}^{j^+}/\tau) + \sum_{k \in \mathcal{I}_{\mathcal{B}} \backslash j, j^+} \exp(\mathbf{h}^j \diamond \mathbf{h}^k/\tau)} \right.$$
$$\left. + \log \frac{\exp(\mathbf{h}^{j^+} \diamond \mathbf{h}^j/\tau)}{\exp(\mathbf{h}^{j^+} \diamond \mathbf{h}^j/\tau) + \sum_{k \in \mathcal{I}_{\mathcal{B}} \backslash j, j^+} \exp(\mathbf{h}^{j^+} \diamond \mathbf{h}^k/\tau)} \right) .$$

The sequence-level contrastive loss aims to separate each sequence apart by separating its positive representation pairs from those of randomly sampled sequences (in-batch negatives). The token-level and sequence-level can complement each other for two main reasons. First, the sequence-level discrimination can help cross-sequence token-level separation. Second, as semantically irrelevant sequences can still share some similar tokens, better token-level separation can help the sequence-level discrimination be more driven by the semantically irrelevant tokens.

### 3.4 CONTRAGEN

In summary, CONTRAGEN optimizes both the token-level and sequence-level contrastive learning objectives in addition to the standard causal language modeling objective as follows

$$\mathcal{L}_{\text{CONTRAGEN}} = \mathcal{L}_{\text{CLM}} + \mathcal{L}_{\text{CONTRAGEN-TOK}} + \mathcal{L}_{\text{CONTRAGEN-SEQ}} .$$

Unless otherwise specified, we weight each loss in CONTRAGEN equally and set the temperature $\tau = 0.05$ for both contrastive losses. Although better performance can be achieved by hyperparameter optimization, we mainly investigate how CONTRAGEN improves the representation quality and boost the zero-shot transfer learning performance on both discrimination and generation tasks. We therefore leave hyperparameter optimization in a supervised setting for future work.

**Positive pair of representations** We consider the simple yet effective dropout-based augmentation (Gao et al., 2021), where the representation pairs are obtained by forwarding each sequence twice. Unlike the existing findings that the dropout-based augmentation can boost the contrastive learning performance when (continually) training a language model, we find that the trends can be different when evaluating on the discrimination tasks and the generation tasks. We discuss with detailed ablation study in both Section 4.4 and Appendix D.2.

## 4 EXPERIMENTS

We design experiments in both natural language and programming language tasks to address the four main questions listed in the Introduction section.

### 4.1 DATA AND MODELS

**Data & Models** For text, we continue train GPT-2 (Radford et al., 2019) on WikiText-103, a collection of over 100 million tokens extracted from the set of verified Good and Featured articles on Wikipedia (Merity et al., 2017). For code, we continue train CodeGen 350M monolingual (Nijkamp et al., 2022) on collected permissively licensed Python code from GitHub. Please refer to Appendix A.1 for the training details. We consider the following objectives for continual training both models.

- **CLM**. The standard left to right autoregression objective for training causal language models, which is also the objective used for pretraining both GPT-2 and CodeGen.

- **SimCTG** (Su et al., 2022). A margin-based token-level contrastive learning framework aims to separate tokens at each distinct location within a sequence apart from each other based on a predefined margin value.[2]
- **CONTRAGEN-TOK** & **CONTRAGEN-SEQ**. In addition to our full model CONTRAGEN, we also evaluate its components CONTRAGEN-TOK and CONTRAGEN-SEQ defined in Section 3.4.

## 4.2 EVALUATE ON NATURAL LANGUAGE

We first evaluate our model on discrimination and generation tasks in natural language.

| Model | STS12 | STS13 | STS14 | STS15 | STS16 | SICK-R | STS-B | Avg. |
|---|---|---|---|---|---|---|---|---|
| GPT2 | 25.84 | 28.90 | 26.20 | 34.74 | 35.70 | 42.72 | 26.27 | 31.48 |
| CLM | 27.14 | 20.34 | 18.73 | 37.56 | 27.40 | 35.70 | 27.97 | 27.83 |
| SimCTG | 30.32 | 37.10 | 31.99 | 39.68 | 42.73 | 46.26 | 25.27 | 36.19 |
| CONTRAGEN-TOK | 37.28 | 37.63 | 31.33 | 54.78 | 50.16 | 48.10 | 34.95 | 42.03 |
| CONTRAGEN-SEQ | 29.66 | 39.89 | 34.50 | 43.20 | 41.99 | 44.52 | 25.51 | 37.04 |
| CONTRAGEN | **37.54** | **45.23** | **36.41** | **56.74** | **50.30** | **51.52** | **39.49** | **45.32** |

Table 1: Spearman rank correlation between the cosine similarity of sentence representation pairs and the ground truth similarity scores.

### 4.2.1 SEMANTIC TEXTUAL SIMILARITY

We first evaluate semantic textual similarity (STS), the most commonly used benchmark for evaluating the semantic discrimination capability of representations. STS consists of seven tasks, namely STS 2012-2016 (Agirre et al., 2012; 2013; 2014; 2015; 2016), the STS Benchmark (Cer et al., 2017), and the SICK-Relatedness (Marelli et al., 2014). Human annotators provide a fine-grained similarity score ranging from 0 to 5 for each sequence pair in STS. Same as Reimers & Gurevych (2019), for the sequence pairs in each dataset, we report the overall Spearman's correlation between the cosine similarities of representations and the human-provided similarity scores in Table 1.

**Contrastive Learning Leads to Discriminative Representations** Table 1 shows both GPT-2 and the one continually trained with CLM perform poorly on STS, which is a consequence of the poorly discriminative representations where the cosine similarities between both semantically similar and dissimilar pairs are almost at one (see Figure 4 in Appendix B.1). Also, continuing to train GPT-2 with CLM on WikiText-103 worsens performance. We hypothesize this is because WikiText-103 is much smaller than the original dataset used for GPT-2 pretraining and hence degenerates the transfer learning performance on STS. In contrast, both CONTRAGEN and SimCTG largely outperforms GPT-2, yet still, CONTRAGEN attains 25% relative improvement over SimCTG. Moreover, with the token-level contrastive learning only, CONTRAGEN-TOK still outperforms SimCTG on almost all STS benchmarks and the trend remains the same even without the dropout-based augmentation (see Appendix D.3). Therefore, we posit our temperature-based contrastive learning allows more flexibility in encouraging the semantic-dependent separations among tokens, while requiring a prefixed separation margin between tokens within the same sequence (context) as SimCTG does, is not ideal.

**Sequence-level vs. Token-level Contrastive Learning** Table 1 also indicates that CONTRAGEN-TOK and CONTRAGEN-SEQ complement each other, as CONTRAGEN consistently performs better than them on all STS tasks. Note that, CONTRAGEN-SEQ performs worse than CONTRAGEN-TOK. It is surprising, especially since STS mainly assesses the sequence-level representation quality. We investigate this by dividing the sequence pairs into two groups – semantically similar pairs with human-annotated similarity scores no less than 0.7 and dissimilar pairs with human scores no larger than 0.3.[3] We plot the rank of the model inferred similarity scores against the human similarity scores in Figure 2 (left). As we can see, CONTRAGEN-SEQ struggles in ranking semantically dissimilar sequence pairs higher and similar pairs lower. This suggests that the token-level contrastive loss is essential for making the sequence-level representations robust to spurious patterns of tokens

---

[2]For all experiments in this section, we set the margin $\rho = 0.5$ as recommended in Su et al. (2022).

[3]For better illustration, we scaled the human annotated scores from [0, 5] to [0,1] in Figure 2.

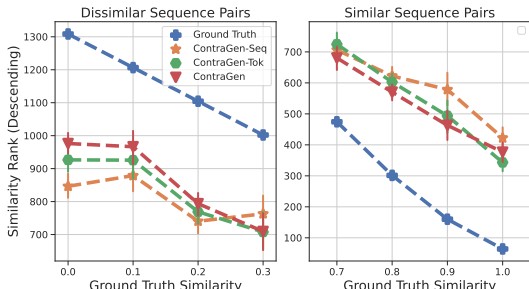

| Similar Sequence Pair | Model | Rank↓ |
|---|---|---|
| **S1:** a woman is stabbing a potato with a fork **S2:** a woman is puncturing a potato with a fork | Ground Truth | 40 |
| | CONTRAGEN-SEQ | 501 |
| | CONTRAGEN-TOK | 272 |
| | CONTRAGEN | **251** |
| **Dissimilar Sequence Pair** | Model | Rank↑ |
| **S1:** a man is opening a box and taking out paper **S2:** a woman is peeling a potato | Ground Truth | 1310 |
| | CONTRAGEN-SEQ | 400 |
| | CONTRAGEN-TOK | 1054 |
| | CONTRAGEN | **1181** |

Figure 2: CONTRAGEN-TOK is essential for making the sequence-level representations robust to spurious patterns of words or phrases (results reported on STS-B).

| Model | Ground Truth Text | | | | Generated Text | | | | | |
|---|---|---|---|---|---|---|---|---|---|---|
| | PPL↓ | ISim | ESim(D) | Disc(D)↑ | MAUVE↑ | ISim | ESim(S) | Disc(S)↓ | ESim(D) | Disc(D)↑ |
| GPT-2 | 47.50 | 0.96 | 0.96 | 0.00 | 0.893 | 0.95 | 0.95 | **0.00** | 0.95 | 0.00 |
| CLM | **22.48** | 0.95 | 0.95 | 0.00 | 0.945 | 0.95 | 0.95 | **0.00** | 0.94 | 0.01 |
| SimCTG | 22.51 | 0.40 | 0.19 | 0.44 | 0.952 | 0.40 | 0.35 | 0.11 | 0.19 | 0.54 |
| CONTRAGEN-TOK | 22.99 | 0.35 | 0.15 | 0.56 | **0.953** | 0.35 | 0.29 | 0.12 | 0.16 | 0.49 |
| CONTRAGEN-SEQ | 22.60 | 0.29 | 0.08 | **0.88** | 0.933 | 0.29 | 0.22 | 0.23 | 0.06 | **0.83** |
| CONTRAGEN | 23.01 | 0.33 | 0.11 | 0.67 | 0.947 | 0.33 | 0.25 | 0.18 | 0.10 | 0.62 |

Table 2: Perplexity and representation quality evaluated on the Wikitext-103 test set. Disc(D) or Esim(D) is computed among two randomly sampled sequences. Disc(S) is computed between the ground truth text and the generated one associated with the same prompt. A lower Disc(S) can indicate better coherence with the ground truth. All metric are reported over the entire test set.

or phrases, *e.g.,* ranking semantically similar sequences with different synonyms low and ranking dissimilar sequences high even in presence of the same phrase (see Figure 2 (right)).

### 4.2.2 TEXT GENERATION

Next, we assess the open-ended language generation capability, where each model is required to generate text continuations given the prefixes from the WikiText-103 test set. Following Su et al. (2022), we set the lengths of prefix and continuation to 32 and 128, respectively. We use nucleus sampling (Holtzman et al., 2020) with top-$p = 0.95$. We report the model perplexity (PPL) in Table 2. PPL evaluates the model prediction confidence on the ground truth continuations in our case. The lower the PPL value, the higher the prediction probabilities assigned to the human-written text. Additionally, we propose the following for evaluating representation quality.

- **Intra-Similarity (ISim)** is the average cosine similarity between the representations of a sequence and its constituent tokens. In this paper, the sequence representation is obtained via mean pooling.

- **Inter-Similarity (ESim)** is the average cosine similarity between the representations of a sequence and each token of another sequence. Regarding contextual representations, it is desired that tokens under **S**imilar contexts (sequences) are pulled closer and hence attaining higher **ESim(S)** score, while tokens from Different contexts are separated apart with lower **ESim(D)** score.

- **Discrimination (Disc)** is defined as Disc = 1 - ESim / ISim where **Disc(S)** and **Disc(D)** associate with ESim(S) and ESim(D) defined above, respectively. Thereby, a lower discrimination score (Disc(S)) between each ground truth and generation is desired for better semantic coherence, while higher discrimination scores (Disc(D)) are desired for generations of randomly sampled prompts for more diverse generations and better isotropic representations.

**Contrastive Learning Trades Off between Model Perplexity and Representation Quality** As reported in Table 2, despite a slight increase in PPL, contrastive learning improves MAUVE (Pillutla et al., 2021) and the representations with better isotropy and discrimination. Furthermore, CONTRA-GEN attains better semantic coherence between the ground truth and generation as indicated by the

| Model | Ruby | | | Python | | | Java | | | Avg. |
|---|---|---|---|---|---|---|---|---|---|---|
| | Ruby | Python | Java | Ruby | Python | Java | Ruby | Python | Java | |
| CodeGen | 16.18 | 5.90 | 0.52 | 2.66 | 18.11 | 0.36 | 1.61 | 1.65 | 10.16 | 6.35 |
| CLM | 16.36 | 6.67 | 0.80 | 3.07 | 15.72 | 0.46 | 1.41 | 2.11 | 10.25 | 6.32 |
| SimCTG | 17.66 | 7.19 | 1.94 | 7.63 | 18.31 | 1.78 | 1.63 | 2.32 | 10.83 | 7.70 |
| CONTRAGEN-TOK | **18.02** | **7.84** | 2.51 | 8.76 | **20.46** | 2.48 | **1.91** | **2.58** | **11.43** | **8.44** |
| CONTRAGEN-SEQ | 16.76 | 5.45 | 1.06 | 7.40 | 16.74 | 1.41 | 1.55 | 2.25 | 10.23 | 6.98 |
| CONTRAGEN | 17.90 | 7.78 | **2.56** | **9.05** | 19.74 | **2.64** | 1.90 | 2.50 | 11.32 | 8.38 |

Table 3: MAP score (%) of zero-shot code search task. The language names mentioned in the top two rows indicate the languages queries and candidates are written in.

lower Disc(S) score and more diverse generations for randomly sampled prompts as indicated by the higher Disc(D) value. We stress that the zero valued Disc(S) score attained by GPT-2 and CLM is indeed a natural consequence of the inferior representations, since the discrimination score between semantically irrelevant sequences is also zero. Also, the slight increase on PPL is probably expected considering PPL is better aligned with the standard CLM objective. Thereby, contrastive learning can be interpreted as regularization that trades-off between PPL and the desired representation properties. We hypothesize CONTRAGEN can improve PPL with new decoding strategy that better leverages the enhanced representations.

## 4.3 EVALUATE ON PROGRAMMING LANGUAGE

In this section, we study the effectiveness of our proposed contrastive learning framework on programming language applications – code search, code completion, and code re-ranking. Different from Section 4.2, CodeGen is pretrained without using dropout from the very beginning. We follow the same setting for all models in this subsection so as to study the effectiveness of CONTRAGEN without the dropout-based augmentation. We also further investigate how dropout would affect the decoder-only models when evaluated on the downstream tasks in Section 4.4 and Appendix D.2.

### 4.3.1 CODE SEARCH

Code search is the task of retrieving relevant code fragments given a code fragment as a *query*. We perform in-language (query and relevant code are in the same language) and cross-language (query and relevant code are in different languages) code search. We provide an example in Figure 5 (Appendix C.1). In this study, we experiment in the *zero-shot* setting - we use the models described in Section 4.1 to generate dense representations of code and perform nearest neighbor search to retrieve relevant code fragments. We use publicly available implementations of Guo et al. (2022)[4].

**Contrastive Learning Yields Discriminative Code Representations** For code-to-code search task, Guo et al. (2022) used problem solutions in Ruby/Python/Java languages from CodeNet (Puri et al., 2021). They propose to use each program as a query and retrieve all programs that solve the same problem. We present a detailed statistics of the dataset in Table 5 (Appendix C.1). We set the maximum sequence length as 512[5] and use cosine similarity between two mean vectors of the last hidden states as relevance scores. We then sort the candidates by their scores to calculate the Mean Average Precision (MAP) score. We present the results for the code search tasks in Table 3.[6]

We observe CONTRAGEN-TOK and CONTRAGEN frameworks improve CodeGen trained with CLM by 33.5% (absolute 2.12) and 32.6% (absolute 2.06) on average, respectively. We also point out that the performance gap between CONTRAGEN-TOK and SimCTG are apples-to-apples comparison since the dropout-based augmentation is not used in either cases. As aforementioned, the consistent better performance of CONTRAGEN-TOK suggests the superiority of our temperature based contrastive learning objective. On the other hand, CONTRAGEN-SEQ improves over the

---

[4]https://github.com/microsoft/CodeBERT/tree/master/UniXcoder/downstream-tasks

[5]We also performed experiments with maximum length 1024 but didn't observe any significant difference.

[6]We present comparison with encoder-only and encoder-decoder models in Table 7 in the Appendix.

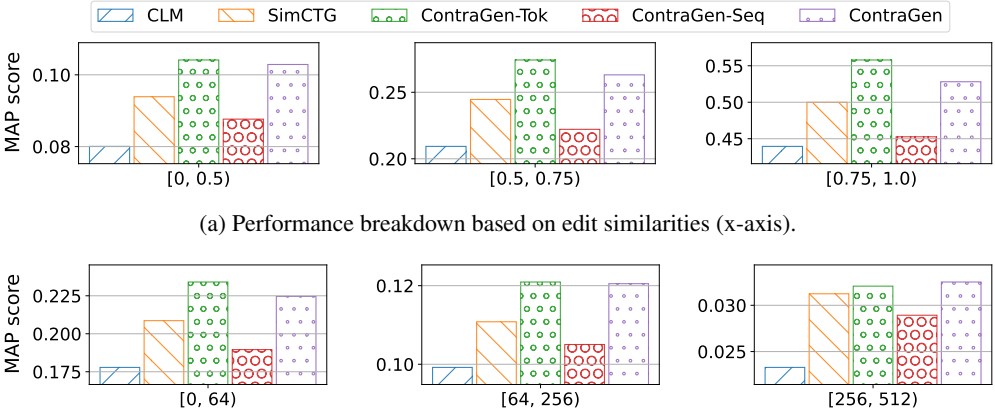

(a) Performance breakdown based on edit similarities (x-axis).

(b) Performance breakdown based on length differences (x-axis).

Figure 3: Code search performances based on (a) and (b) between the query code fragments (in Python) and their relevant code fragments (in Python). We observe that in both cases, CONTRAGEN-TOK outperforms CLM, SimCTG, and CONTRAGEN-SEQ.

CLM baseline by 10.4% only. Code search results indicate that CONTRAGEN-SEQ performs poorly compared to CONTRAGEN-TOK, the performance gap is larger than what we observed in natural language evaluation. We conjecture that CONTRAGEN-TOK generates better discriminative representations for code sequences since the finer-grained understanding of the code tokens is crucial to understanding the code sequences' functionality (semantics). To verify, we check if non-semantic factors impact model performances in the following section.

**Token-level Contrastive Learning is Effective for Code Understanding**   We break down the code search performance based on edit similarities and length differences between query code and their relevant code fragments. While edit similarity indicates how much queries and their relevant code overlap, the length difference indicates whether models effectively capture relevance between two code fragments if they are similar in length or differ significantly. We present the results for Python language in Figure 3 (for all the languages, see Figures 7 & 8 in Appendix C.3). The results show that CONTRAGEN-TOK outperforms CLM, SimCTG, and CONTRAGEN-SEQ irrespective of edit similarities and length differences. Therefore, we can conclude that sequence overlap or length are not the reasons for improvements in CONTRAGEN-TOK. Presumably, a finer-grained understanding of code tokens makes CONTRAGEN-TOK more effective for code representations.

### 4.3.2   CODE COMPLETION AND RE-RANKING

Given a sequence of tokens composed of natural language, function signature, and input-output examples (as a whole, we call them prompt), the goal of the code completion task is to complete the function. To evaluate the functional correctness of a complete code, we use existing benchmarks that include unit tests. If the generated code successfully passes the unit tests, we refer to this as successful execution. We compute pass@$k$ for $k \leq n$ following Chen et al. (2021). In addition, we compare the models in the code re-ranking task - given $n$ sampled code using a code completion model, the goal of code re-ranking is to find an ordering of the code. We use the mean log probability of the sampled code to order them (Chen et al., 2021). For code re-ranking evaluation, we report ranked pass@$k$ (Inala et al., 2022). We present an example in Figure 6 (Appendix C.1) to illustrate the code completion and re-ranking tasks. We detail the evaluation metrics in Appendix C.2.

**Contrastive Learning Improves Source Code Generation**   Chen et al. (2021) introduced HumanEval, a collection of 164 handwritten programming problems and their respective unit tests. Each problem in this dataset is presented using a function signature and supporting docstring, and the task is to complete the body of the function, such that the complete function can pass all unit tests. In all our experiments, we use nucleus sampling (Holtzman et al., 2020) with top $p = 0.95$. We sample $n = 10$ completions per problem with sampling temperature 0.2. Table 4 presents the evaluation results on the HumanEval benchmark.

We see that CONTRAGEN-TOK and CONTRAGEN-SEQ perform comparably to CLM and SimCTG, while CONTRAGEN outperforms them significantly (by 9% and 10.3% in terms of pass@1 accuracy). Similarly, in the code re-ranking task, CONTRAGEN outperforms CLM and SimCTG baselines by 11% and 12% in terms of ranked pass@1 accuracy, respectively. While CONTRAGEN-SEQ underperforms in code completion, it boosts code re-ranking significantly. We hypothesize the improvement is due to the contrastive learning objective aligns with the mean log probability-based re-ranking choice.

| Model | Pass@k | | Ranked Pass@k | |
|---|---|---|---|---|
| | k=1 | k=5 | k=1 | k=5 |
| CodeGen | 12.65 | 16.89 | 13.42 $_{(+0.77)}$ | 17.07 $_{(+0.18)}$ |
| CLM | 13.42 | 18.08 | 15.38 $_{(+1.96)}$ | 18.29 $_{(+0.21)}$ |
| SimCTG | 13.26 | 17.29 | 15.24 $_{(+1.98)}$ | 18.29 $_{(\textbf{+1.00})}$ |
| CONTRAGEN-TOK | 12.96 | 17.01 | 15.24 $_{(+2.96)}$ | 17.68 $_{(+0.67)}$ |
| CONTRAGEN-SEQ | 13.64 | 15.85 | 16.99 $_{(\textbf{+3.35})}$ | 16.46 $_{(+0.61)}$ |
| CONTRAGEN | **14.63** | **18.83** | **17.07** $_{(+2.44)}$ | **18.90** $_{(+0.07)}$ |

Table 4: Evaluation results on the HumanEval benchmark. The numbers in the subscript indicates the difference between ranked pass@k and pass@k. While CONTRAGEN-TOK and CONTRAGEN-SEQ perform competitively to the baselines, CONTRAGEN significantly outperforms them.

## 4.4 DISCUSSION

**Impact of Dropout** Dropout-based augmentation (Gao et al., 2021) has proven to be effective for contrastive learning on language models, which often leads to significant improvement when evaluating on discriminative tasks. We observe the same trend on both GPT-2 and CodeGen (see Tables 8b & 8a in Appendix D.2). However, we observed the opposite when evaluating on language generation, no matter when training with CLM only or together with contrastive learning (see Appendix D.2). Dropout has been one of the key ingredients for training large models, further investigation on the proper ways to use and evaluate it is indeed required. On the bright side, even without dropout, Section 4.3 shows CONTRAGEN still consistently yield considerable improvement.

**Bridge the Gap between Causal and Bidirectional Attention Models on Discriminative Tasks** In comparison with the causal (left-to-right) attention mechanism of the decoder-only models, the bidirectional attention mechanism allows to better leverage the context of the sequence and hence yield better representations for discrimination tasks. Take the encoder-only model as an example, as Table 6 in Appendix shows, both BERT-Base (Devlin et al., 2019) and RoBERTa-Base (Liu et al., 2019) outperform GPT-2 by at least 60% relative performance on STS. Despite the performance gap between CodeGen and the encoder-only or encoder-decoder models decreases in Table 7, it is still significant considering that both the model size and pretraining data size used by CodeGen are much larger. Such large performance gap severely limits the decoder-only models being used in many discriminative tasks. On the bright side, contrastive learning shows the promise to bridge the gap. As seen in Table 6, on STS, CONTRAGEN reduce the relative performance gap from 67.24% (absolute 21.12%) to 16.17% (absolute 7.33%) regarding BERT-Base, and from 84.62% (absolute 26.64%) to 28.24% (absolute 12.8%) regarding RoBERTa-Base. Similarly, Table 7 shows that CONTRAGEN outperforms encoder-decoder models and perform comparably to encoder-only model GraphCodeBERT. Please refer to Appendix D.1 for more detailed discussions.

## 5 CONCLUSION

In this paper, we present CONTRAGEN, an effective contrastive learning framework to resolve the representation degeneration issue of causal language models trained with the standard autoregression objective. We assess the effectiveness of the proposed strategy on a variety of downstream tasks in both natural language domain and programming language domain, where we attain significant improvement on both discrimination tasks and language generation tasks. Moreover, we conduct in-depth analyses on our proposed token-level and sequence-level contrastive losses. Although we only explored on the decoder-only causal language models, our proposed contrastive learning framework can serve as a drop-in term for encoder-decoder, encoder-only, or prefixLM. We leave the explorations as future work.

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

# A  Training Details

## A.1  Setup Details

**Training Data**    For text, we use WikiText-103, a collection of over 100 million tokens extracted from the set of verified Good and Featured articles on Wikipedia (Merity et al., 2017). For code, we collected permissively licensed Python code from GitHub. Following Chen et al. (2021); Nijkamp et al. (2022), we perform filtering and deduplication and further remove data that contains a significant use of non-English languages or is not parsable, resulting in a dataset of 101GB code.

**Model**    We use GPT-2 (Radford et al., 2019) and CodeGen 350M monolingual (Nijkamp et al., 2022) for all experiments on natural language (text) and programming language (code), respectively. We set the batch size to 512 and continue to train GPT-2 on WikiText-103 and CodeGen on the GitHub data for 12 and 2 epochs, respectively. We trained both models using a max sequence length of 512 tokens and 1024 for WikiText-103 and Code data, respectively. We set the learning rate to 2e-5, warm-up steps as 500 with linear annealing after peak learning rate, weight decay of 0.1, the temperature of 0.05 (when using contrastive losses), and gradient clipping of 1.0. We use AdamW optimizer (Loshchilov & Hutter, 2019) with $\beta_1 = 0.9$, $\beta_2 = 0.999$, and $\epsilon = 10^{-8}$ following (Nijkamp et al., 2022). Our training pipeline is based on PyTorch Lightning[7] and we use DeepSpeed (Rasley et al., 2020) for training optimization.

**Processing Code Training Data**    Our preprocessing strategy for code datasets used for training is designed to ensure that we optimize for data utilization while retaining the syntactic structure of programming language sequences. We also eliminate duplicate sequences since this benefits training large language models (Lee et al., 2022). Specifically, we break long sequences into chunked sequences of smaller lengths to retain most parts of the original program. Further, we maintain syntactic structure in the chunks by ensuring that each chunk ends with a '\n' character. Each chunk obtained this way contains at most `max_chars_per_seq` characters where `max_chars_per_seq` = `max_tokens_per_seq` * `chars_per_tok`. In our experiments, we fix `chars_per_tok` = 3.2 and `max_tokens_per_seq` = 1024. We also perform deduplication using character-based exact matches between chunked sequences over the entire dataset. This step helps eliminate exact-duplicates that might be present after the chunking stage.

# B  More on Natural Language Evaluation

## B.1  Representation Quality Evaluation on STS

In this section, we dive deep into the model performance on the STS benchmarks. For each sequence pair in STS, a fine-grained similarity score ranging from 0 to 5 is provided, with a high similarity score indicating semantically similar pairs and low similarity scores suggesting semantically dissimilar or irrelevant pairs. For better illustration, we scale the human-annotated similarity scores to $[0, 1]$ to align with the model-predicted cosine similarity scores. This does not affect the evaluation as the spearman correlation reported in Section 4.2 is a rank-based correlation metric.

**CLM yields poorly discriminative representations**    We report the model predicted similarity scores of sequence pairs in Figure 4a. A good model is expected to yield representations that attain higher similar scores between similar sequence pairs and lower similarity values for dissimilar sequences. A large gap between the similarity scores of similar and dissimilar pairs is desired. However, as seen in the left column of Figure 4, the similarity scores attained by the model trained with the standard CLM only objective are almost at one for both similar sequence pairs and dissimilar pairs. This again suggests that the representations yielded by CLM are squeezed into a tiny cone in the representation space rather than being scattered apart to leverage the vector space's capacity better. Despite the resulting similarity ranks are not entirely flattened, as shown in the right column in Figure 4b, CLM struggles in ranking similar sequences lower and dissimilar sequences higher as a consequence of its poorly discriminative representations such that low similarity scores can be assigned to the semantically similar and high similarity scores to dissimilar sequence pairs.

---

[7]https://www.pytorchlightning.ai/

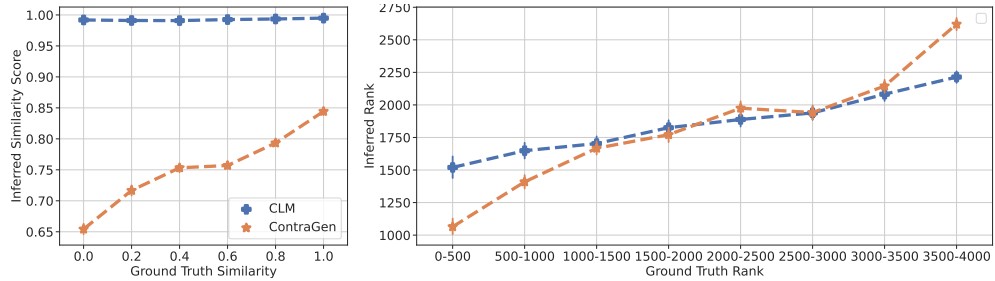

(a) STS14: **(Left)** Predicted cosine similarity vs. human annotated ground truth. **Right** Similarity ranking according to the model predicted similarity scores vs Human similarity based ranking.

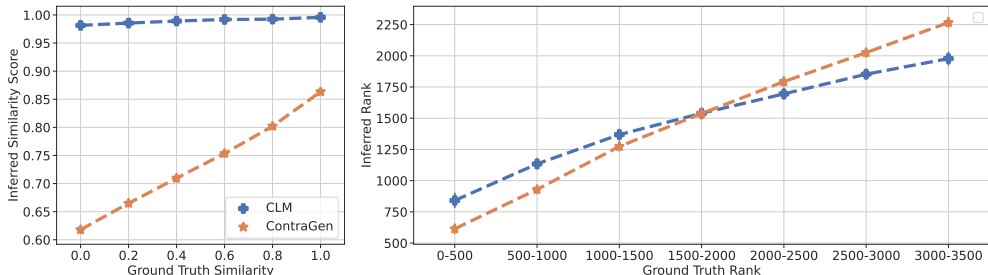

(b) STS15: **(Left)** Predicted cosine similarity vs. human annotated ground truth. **Right** Similarity ranking according to the model predicted similarity scores vs Human similarity based ranking.

Figure 4: CLM versus Contrastive Learning in Similarity Prediction and Ranking. We report the results on two STS benchmarks: (1) *STS14* where CLM performs the worst compared to its performance on the other STS tasks; and *STS15* where CLM attains the best performance when comparing with its own. For the purpose of illustration, we scale the human annotated similarity scores from $[0, 5]$ to $[0, 1]$. A good language model is expected to predict discriminative similarity scores such that the resulting ranking results are as close as the ranks provided by human as possible.

**CONTRAGEN leads to representations with better discrimination and contextulization** In comparison, Figure 4a also validates that contrastive learning effectively yields more discriminative representations with a comparatively larger similarity gap between the similar pairs and dissimilar pairs. Thereby, the similarity ranking results of the sequence pairs are more aligned with those obtained according to similarity scores provided by a human. Figure 4b shows that CONTRAGEN consistently outperforms the other models, with the ranking results better matching the ground truth.

## C    MORE ON PROGRAMMING LANGUAGE EVALUATION

### C.1    TASKS EXAMPLES AND STATISTICS

In Figure 5, we present an example of query code fragment in Python and relevant code fragments in Python and Java, respectively. While in-language code-to-code search refers to retrieving relevant code fragments in the same language, cross-language code-to-code search refers to retrieving code fragments in a different language. We present the statistics of the code search dataset in Table 5. To demonstrate the code completion task, we illustrate an example in Figure 6.

### C.2    EVALUATION METRICS

**Mean Average Precision (MAP)**    For a set of queries, it indicates the mean of the average precision scores for each query.

$$MAP = \frac{\sum_{q=1}^{Q} AveP(q)}{Q}$$

where $Q$ is the number of queries.

|  | Ruby | Python | Java |
|---|---|---|---|
| Total problems | 1,708 | 2072 | 3142 |
| Total #solution | 11,744 | 15,594 | 23,530 |
| Avg. #solution / problem | 6.9 | 7.5 | 7.5 |
| Avg. length / solution | 160.4 | 214.4 | 894.9 |
| Stdev. of length / solution (problem-wise) | 112.80 | 113.79 | 813.0 |
| Solutions with length $> 512$ | 409 | 1,200 | 10,023 |
| Solutions with length $> 1024$ | 78 | 278 | 4,766 |
| Avg. edit similarity | $0.48_{(+0.13)}$ | $0.52_{(+0.13)}$ | $0.49_{(+0.13)}$ |

Table 5: Statistics of code-to-code search task dataset created from CodeNet (Puri et al., 2021). We truncate the code if its length exceeds maximum sequence length which is set to 512.

**Pass@k**   Given a problem (code prompt as as shown in Figure 6), pass@k indicates functional correctness of model generated code samples. A problem is considered solved if any sample passes the unit tests. Following Chen et al. (2021), we generate $n \geq k$ samples per problem (in this paper, we use $n = 10$ and $k \in \{1, 5\}$), count the number of correct samples $c \leq n$ that pass unit tests, and calculate the unbiased estimator of pass@$k$ as:

$$\text{pass@}k := \underset{Problems}{\mathbb{E}} \left[ 1 - \frac{\binom{n-c}{k}}{\binom{n}{k}} \right] .$$

**Ranked Pass@k**   Unlike Pass@k, where we randomly chose $k$ out of $n$ samples, in ranked pass@k, we chose the top-$k$ samples based on model-provided scores and then compute pass@k.

### C.3   Detailed Code Search Results

We provide comparison between encoder-only (Feng et al., 2020; Guo et al., 2021), encoder-decoder (Ahmad et al., 2021; Wang et al., 2021), and decoder-only models (main focus of this work) on the zero-shot code-to-code search task in Table 7. We see that CONTRAGEN-TOK and CONTRAGEN outperforms the encoder-only model CodeBERT and both the encoder-decoder models. It is important to note that the comparison across these models is not apple-to-apple as these models differ in size, the scale of pretraining, and language settings. This comparison's purpose is to show the promise of decoder-only models being used in discriminative tasks like code search.

We further break down the code search performances based on edit similarities and length differences between query code and their relevant code fragments. We present the results in Figure 7 and 8. We observe a similar performance trend in all three languages, although cross-lingual search performance still needs to improve. Nonetheless, the objective of this performance analysis is to show that sequence overlap or length are not the reasons for improvements in CONTRAGEN-TOK. Instead, a finer-grained understanding of code tokens due to the token-level contrastive learning makes CONTRAGEN-TOK more effective for code representations.

## D   More Analysis and Discussions

### D.1   Contrastive Learning Bridge the Gap Between Causal and Bidirectional Attention Models on Discriminative Tasks

Compared to the causal (left-to-right) attention mechanism of the decoder-only models, the bidirectional attention mechanism in both encoder-only and encoder-decoder models allows to better leverage the context of the sequence. It hence leads to better representations when evaluated on discrimination tasks.

Taking the encoder-only models as an example. As shown in Table 6, on average, BERT-Base (Devlin et al., 2019) and Roberta-Base (Liu et al., 2019) outperform GPT-2 with 67.24% (absolute 21.12%) and 84.62% (absolute 26.64%) relative improvement on STS, respectively. Although

the performance gap between CodeGen and the BERT models trained on programming languages (CodeBERT (Feng et al., 2020), GraphCodeBERT (Guo et al., 2021)) decreases or even diminishes when evaluated on the Code-Search tasks, the performance gap is still significant because both the model size and pretraining data in CodeGen are much larger than those by the encoder-only models in Table 7. Similar trends were observed regarding the performance gap between the decoder-only and encoder-decoder models on both natural language (Lewis et al., 2020; Raffel et al., 2020) and programming language (Ahmad et al., 2021; Wang et al., 2021).

| Model | STS12 | STS13 | STS14 | STS15 | STS16 | SICK-R | STS-B | Avg. |
|---|---|---|---|---|---|---|---|---|
| Encoder-only Models | | | | | | | | |
| BERT-Base | 30.92 | 59.96 | 47.72 | 60.35 | 63.72 | 58.25 | 47.36 | 52.65 |
| RoBERTa-Base | 53.95 | 47.42 | 55.87 | 64.73 | 63.55 | 62.94 | 58.40 | 58.12 |
| Encoder-Decoder Models | | | | | | | | |
| BART-Base | 34.46 | 52.49 | 44.50 | 62.51 | 61.99 | 57.72 | 52.30 | 52.28 |
| T5-Base | 37.78 | 56.81 | 49.37 | 65.50 | 64.65 | 60.11 | 57.52 | 55.96 |
| Decoder-only Models | | | | | | | | |
| GPT2 | 25.84 | 28.90 | 26.20 | 34.74 | 35.70 | 42.72 | 26.27 | 31.48 |
| CLM | 27.14 | 20.34 | 18.73 | 37.56 | 27.40 | 35.70 | 27.97 | 27.83 |
| SimCTG | 30.32 | 37.10 | 31.99 | 39.68 | 42.73 | 46.26 | 25.27 | 36.19 |
| CONTRAGEN-TOK | 37.28 | 37.63 | 31.33 | 54.78 | 50.16 | 48.10 | 34.95 | 42.03 |
| CONTRAGEN-SEQ | 29.66 | 39.89 | 34.50 | 43.20 | 41.99 | 44.52 | 25.51 | 37.04 |
| CONTRAGEN | **37.54** | **45.23** | **36.41** | **56.74** | **50.30** | **51.52** | **39.49** | **45.32** |

Table 6: Spearman rank correlation between the cosine similarity of sentence representation pairs and the ground truth similarity scores.

| Model | Ruby | | | Python | | | Java | | | Avg. |
|---|---|---|---|---|---|---|---|---|---|---|
| | Ruby | Python | Java | Ruby | Python | Java | Ruby | Python | Java | |
| Encoder-only Models | | | | | | | | | | |
| CodeBERT | 13.55 | 3.18 | 0.71 | 3.12 | 14.39 | 0.96 | 0.55 | 0.42 | 7.62 | 4.94 |
| GraphCodeBERT | 17.01 | 9.29 | 6.38 | 5.01 | 19.34 | 6.92 | 1.77 | 3.50 | 13.31 | 9.17 |
| Encoder-Decoder Models | | | | | | | | | | |
| PLBART | 18.60 | 10.76 | 1.90 | 8.27 | 19.55 | 1.98 | 1.47 | 1.27 | 10.41 | 8.25 |
| CodeT5-base | 18.22 | 10.02 | 1.81 | 8.74 | 17.83 | 1.58 | 1.13 | 0.81 | 10.18 | 7.81 |
| Decoder-only Models | | | | | | | | | | |
| CodeGen | 16.18 | 5.90 | 0.52 | 2.66 | 18.11 | 0.36 | 1.61 | 1.65 | 10.16 | 6.35 |
| CLM | 16.36 | 6.67 | 0.80 | 3.07 | 15.72 | 0.46 | 1.41 | 2.11 | 10.25 | 6.32 |
| SimCTG | 17.66 | 7.19 | 1.94 | 7.63 | 18.31 | 1.78 | 1.63 | 2.32 | 10.83 | 7.70 |
| CONTRAGEN-TOK | 18.02 | 7.84 | 2.51 | 8.76 | 20.46 | 2.48 | 1.91 | 2.58 | 11.43 | 8.44 |
| CONTRAGEN-SEQ | 16.76 | 5.45 | 1.06 | 7.40 | 16.74 | 1.41 | 1.55 | 2.25 | 10.23 | 6.98 |
| CONTRAGEN | 17.90 | 7.78 | 2.56 | 9.05 | 19.74 | 2.64 | 1.90 | 2.50 | 11.32 | 8.38 |

Table 7: MAP score (%) of zero-shot code-to-code search task. The language names mentioned in the top two rows indicate the languages queries and candidates are written in.

The large performance gap severely limits the decoder-only models used in many discriminative tasks. To this end, contrastive learning shows the promise to largely bridge the gap. As seen in Table 6, on STS, CONTRAGEN reduce the relative performance gap from 67.24% (absolute 21.12%) to 16.17% (absolute 7.33%) regarding BERT-Base, and from 84.62% (absolute 26.64%) to 28.24% (absolute 12.8%). Similarly, Table 7 shows that CONTRAGEN outperfors encoder-decoder models and perform comparably to encoder-only model GraphCodeBERT.

| Model | Ruby | | | Python | | | Java | | | Avg. |
|---|---|---|---|---|---|---|---|---|---|---|
| | Ruby | Python | Java | Ruby | Python | Java | Ruby | Python | Java | |
| CLM $_{+\text{Dropout}}$ | 18.04 | 6.47 | 1.21 | 5.52 | 18.70 | 1.18 | 1.62 | 2.35 | 11.26 | 7.37 |
| CLM $_{-\text{Dropout}}$ | 16.36 | 6.67 | 0.8 | 3.07 | 15.72 | 0.46 | 1.41 | 2.11 | 10.25 | 6.32 |
| CONTRAGEN $_{+\text{Dropout}}$ | **20.09** | **8.84** | **3.66** | **9.25** | **22.39** | **3.13** | **1.93** | **3.06** | **12.02** | **9.37** |
| CONTRAGEN $_{-\text{Dropout}}$ | 17.98 | 7.57 | 2.51 | 8.94 | 20.25 | 2.55 | 1.92 | 2.57 | 11.34 | 8.40 |

(a) MAP score (%) of zero-shot code-to-code search.

| Model | STS12 | STS13 | STS14 | STS15 | STS16 | SICK-R | STS-B | Avg. |
|---|---|---|---|---|---|---|---|---|
| CLM $_{+\text{Dropout}}$ | 27.14 | 20.34 | 18.73 | 37.56 | 27.40 | 35.70 | 27.97 | 27.83 |
| CLM $_{-\text{Dropout}}$ | 25.60 | 15.23 | 13.95 | 31.64 | 28.13 | 34.96 | 26.15 | 25.09 |
| CONTRAGEN $_{+\text{Dropout}}$ | 37.54 | **45.23** | **36.41** | **56.74** | **50.30** | **51.52** | **39.49** | **45.32** |
| CONTRAGEN $_{-\text{Dropout}}$ | **38.22** | 40.15 | 33.57 | 53.16 | 45.35 | 47.47 | 36.10 | 42.00 |

(b) Spearman rank correlations between the cosine similarity of sentence representation pairs and the ground truth similarity scores for STS benchmarks.

Table 8: Discriminative task performances with ($_{+\text{Dropout}}$) and without ($_{-\text{Dropout}}$) Dropout augmentation applied to CLM and CONTRAGEN. We apply Dropout (0.1) to all the layers of the models.

## D.2 DROPOUT FOR CONTRASTIVE LEARNING WITH DECODER-ONLY MODEL

Gao et al. (2021) showed that dropout-based augmentation is an effective strategy for unsupervised contrastive learning and followup works (Chuang et al., 2022; Wu et al., 2022) endorse the effectiveness. This motivates us to study dropout-based augmentation in our proposed contrastive learning framework. We present the results on discriminative and generation tasks in Table 8 and 9, respectively. From results, it is evident that the adoption of dropout-based augmentation improves the discrimination task performances, which corroborates the findings of Gao et al. (2021).

On the other hand, dropout-based augmentation hurts the generation task performances. While for code completion, we anticipated that dropout-based augmentation would hurt the performances since we use CodeGen (Nijkamp et al., 2022) in this work which does not use dropout during pretraining. However, to our surprise, we observe a drop in perplexity due to the dropout-based augmentation that does not go with our anticipation (unlike CodeGen, GPT-2 is pretrained with dropout). Therefore, we leave this as a future exploration to dive deeper into the reasoning behind this finding.

| Model | Pass@k | | Ranked Pass@k | |
|---|---|---|---|---|
| | k=1 | k=5 | k=1 | k=5 |
| CLM $_{+\text{Dropout}}$ | 13.19 | 15.92 | 13.41 $_{(+0.22)}$ | 16.46 $_{(+0.54)}$ |
| CLM $_{-\text{Dropout}}$ | 13.42 | 18.08 | 15.38 $_{(+1.96)}$ | 18.29 $_{(+0.21)}$ |
| CONTRAGEN $_{+\text{Dropout}}$ | 13.19 | 15.92 | 13.41 $_{(+0.22)}$ | 16.46 $_{(+3.05)}$ |
| CONTRAGEN $_{-\text{Dropout}}$ | 13.41 | 17.89 | 15.24 $_{(+1.83)}$ | 18.90 $_{(+1.01)}$ |

(a) Evaluation results on the HumanEval benchmark. The numbers in the subscript indicates the difference between ranked pass@k and pass@k accuracy.

| | CLM $_{-\text{Dropout}}$ | CLM $_{+\text{Dropout}}$ | CONTRAGEN $_{-\text{Dropout}}$ | CONTRAGEN $_{+\text{Dropout}}$ |
|---|---|---|---|---|
| Perplexity | 21.86 | 22.48 | 22.07 | 23.01 |

(b) Perplexity of continually trained GPT-2 on the test set of WikiText-103.

Table 9: Generation task performances with ($_{+\text{Dropout}}$) and without ($_{-\text{Dropout}}$) Dropout augmentation applied to CLM and CONTRAGEN. We apply Dropout (0.1) to all the layers of the models.

### D.3 CONTRAGEN CONSISTENTLY OUTPERFORM SIMCTG

To better understand the performance gap between CONTRAGEN and SimCTG (Su et al., 2022), we run the following ablations on GPT-2 and report the evaluations on STS.

In Table 10, we report the results of (1) running CONTRAGEN w/o dropout-based data augmentation and compare it with the original SimCTG model; and (2) augmenting SimCTG with both the sequence-level contrastive loss and dropout-based augmentation and compare it with our proposed CONTRAGEN model. As we can see, CONTRAGEN consistently outperforms SimCTG. Figure 10 together with our results reported in Section 4.3 where we disabled the dropout-based augmentation for CONTRAGEN and its variations, but still observed consistent better performance than SimCTG on both discrimination and generation tasks, conclude that CONTRAGEN is better than SimCTG across domains and settings.

| Model | STS12 | STS13 | STS14 | STS15 | STS16 | SICK-R | STS-B | Avg. |
|---|---|---|---|---|---|---|---|---|
| SimCTG | 30.32 | 37.10 | 31.99 | 39.68 | 42.73 | 46.26 | 25.27 | 36.19 |
| CONTRAGEN $_{-\text{Dropout}}$ | 38.22 | 40.15 | 33.57 | 53.16 | 45.35 | 47.47 | 36.10 | 42.00 |
| SimCTG $_{+\mathcal{L}_{\text{Seq}}+\text{Dropout}}$ | **38.70** | 43.60 | 36.29 | 50.01 | 45.19 | 48.25 | 33.36 | 42.20 |
| CONTRAGEN $_{+\text{Dropout}}$ | 37.54 | **45.23** | **36.41** | **56.74** | **50.30** | **51.52** | **39.49** | **45.32** |

Table 10: CONTRAGEN consistently outperform SimCTG (Su et al., 2022) even without dropout based data augmentation (first two rows); or augmenting SimCTG with dropout and sequence-level contrastive loss defined in Equation 1.

Query Program in Python

```python
1  import math
2  h,w=map(int, input().split())
3  if h%3==0 or w%3==0:
4      print(0)
5  else:
6      x,y=max(h,w),min(h,w)
7      ans=y
8      for hi in range(1,h):
9          M=max(hi*w,(h-hi)*((w+1)//2),(h-hi)*(w//2))
10         m=min(hi*w,(h-hi)*((w+1)//2),(h-hi)*(w//2))
11         ans=min(ans,M-m)
12     for wi in range(1,w):
13         M=max(wi*h,(w-wi)*((h+1)//2),(w-wi)*(h//2))
14         m=min(wi*h,(w-wi)*((h+1)//2),(w-wi)*(h//2))
15         ans=min(ans,M-m)
16     print(ans)
```

Relevant Program in Python

```python
1  def solve(H,W):
2      p1 = [H//3*W, (H-H//3)*(W//2), (H-H//3)*(W-W//2)]
3      p2 = [ceil(H/3)*W, (H-ceil(H/3))*(W//2), (H-ceil(H/3))*(W-W//2)]
4      S1 = max(p1)-min(p1)
5      S2 = max(p2)-min(p2)
6      S3 = 0 if H%3==0 else W
7      return min(S1,S2,S3)
8
9  from math import ceil
10 H, W = map(int, input().split())
11 print(min(solve(H,W), solve(W,H)))
```

Relevant Program in Java

```java
1  import java.util.*;
2  public class Main {
3    public static void main(String[] args){
4      Scanner sc = new Scanner(System.in);
5      long w = sc.nextInt();
6      long h = sc.nextInt();
7      if(w%3==0 || h%3==0)
8        System.out.println(0);
9      else
10       System.out.println(Math.min(solve(w, h), solve(h, w)));
11   }
12   static long solve(long w, long h){
13     long min = Long.MAX_VALUE;
14     for(int i=1;i<h;i++){
15       long a = w*i;
16       long b, c = 0, 0;
17       if(w%2==0){
18         b = w/2*(h-i);
19         c = b;
20         min = Math.min(min, Math.max(a, Math.max(b, c))-Math.min(a, Math.min(b, c)));
21       }
22       else if((h-i)%2==0){
23         b = w*((h-i)/2);
24         c = b;
25         min = Math.min(min, Math.max(a, Math.max(b, c))-Math.min(a, Math.min(b, c)));
26       }
27       else{
28         b = w*((h-i)/2);
29         c = w*((h-i)/2+1);
30         min = Math.min(min, Math.max(a, Math.max(b, c))-Math.min(a, Math.min(b, c)));
31         b = w/2*(h-i);
32         c = (w/2+1)*(h-i);
33         min = Math.min(min, Math.max(a, Math.max(b, c))-Math.min(a, Math.min(b, c)));
34       }
35     }
36     return min;
37   }
38 }
```

Figure 5: An example of a query and relevant documents in code-to-code search task, where both query and candidates are complete programs that solve a programming problem. In this search task, each program is used as a query and retrieve all programs (from a collection of 11,744/15,594/23,530 programs in Ruby, Python, and Java, respectively) that solve the same problem.

---

Prompt (function signature and docstring)

```
1  from typing import List
2
3
4  def has_close_elements(numbers: List[float], threshold: float) -> bool:
5      """ Check if in given list of numbers, are any two numbers closer to each other than
6      given threshold.
7      >>> has_close_elements([1.0, 2.0, 3.0], 0.5)
8      False
9      >>> has_close_elements([1.0, 2.8, 3.0, 4.0, 5.0, 2.0], 0.3)
10     True
11     """
```

---

Unit tests

```
1  def check(candidate):
2      assert candidate([1.0, 2.0, 3.9, 4.0, 5.0, 2.2], 0.3) == True
3      assert candidate([1.0, 2.0, 3.9, 4.0, 5.0, 2.2], 0.05) == False
4      assert candidate([1.0, 2.0, 5.9, 4.0, 5.0], 0.95) == True
5      assert candidate([1.0, 2.0, 5.9, 4.0, 5.0], 0.8) == False
6      assert candidate([1.0, 2.0, 3.0, 4.0, 5.0, 2.0], 0.1) == True
7      assert candidate([1.1, 2.2, 3.1, 4.1, 5.1], 1.0) == True
8      assert candidate([1.1, 2.2, 3.1, 4.1, 5.1], 0.5) == False
9
10 check(has_close_elements)
```

---

Completion 1 (passed; mean_logp: -0.1146)

```
1      for i in range(len(numbers) - 1):
2          for j in range(i + 1, len(numbers)):
3              if abs(numbers[i] - numbers[j]) < threshold:
4                  return True
5      return False
```

---

Completion 2 (failed; mean_logp: -0.1211)

```
1      for i in range(len(numbers) - 1):
2          for j in range(i + 1, len(numbers)):
3              if abs(numbers[i] - numbers[j]) > threshold:
4                  return True
5      return False
```

---

Completion 3 (failed; mean_logp: -0.1598)

```
1      for i in range(len(numbers) - 1):
2          if abs(numbers[i] - numbers[i + 1]) > threshold:
3              return True
4      return False
```

---

Completion 4 (failed; mean_logp: -0.1607)

```
1      for i in range(len(numbers) - 1):
2          if abs(numbers[i] - numbers[i + 1]) > threshold:
3              return False
4      return True
```

---

Completion 5 (failed; mean_logp: -0.3904)

```
1      return abs(sum(numbers) - sum(numbers) / len(numbers)) < threshold
```

---

Figure 6: An example of prompt from HumanEval and five unique code completions generated by the CONTRAGEN model. We rank them based on the mean_logp scores. Considering these 5 completions and one of the passes unit tests, pass@1 is 0.2 while ranked pass@1 is 1.0.

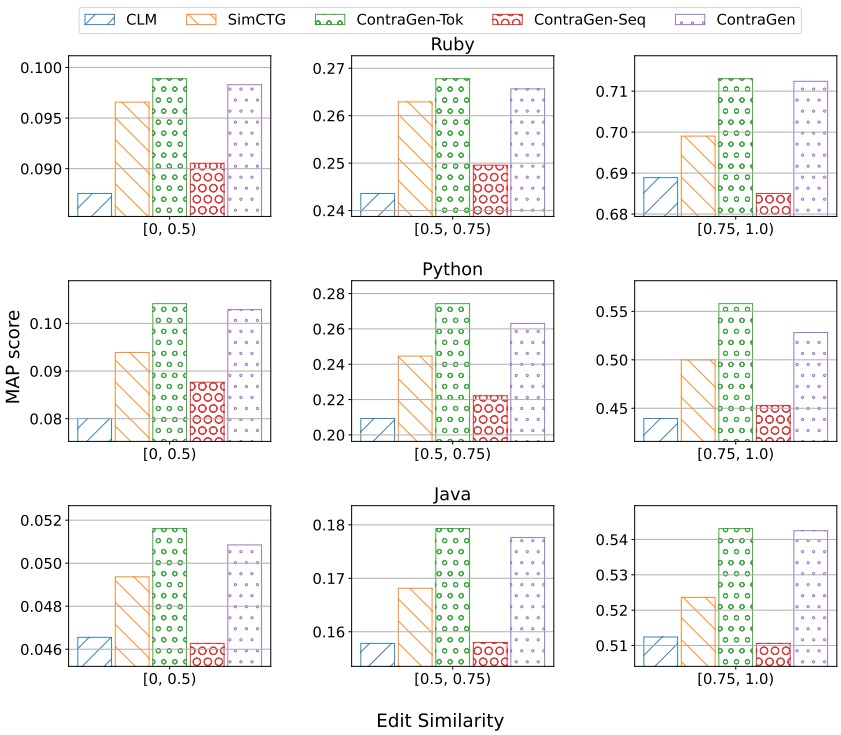

Figure 7: Code-to-code search performance (MAP score) breakdown for different models based on the edit similarities ([0, 1]) between the query code fragments and the relevant code fragments. Higher and lower edit similarity indicates the search task is trivial or difficult, respectively.

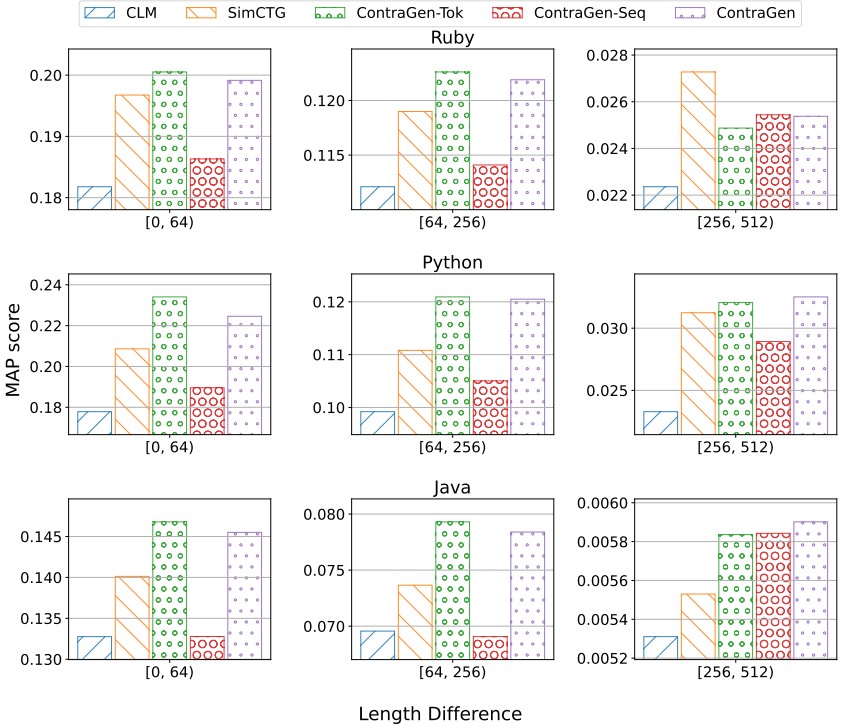

Figure 8: Code-to-code search performance (MAP score) breakdown based on (absolute) length differences between the query code fragments and their relevant code fragments.

