# OpenReview forum: "ContraGen: Effective Contrastive Learning For Causal Language Model"
_ICLR.cc/2023/Conference — Submitted to ICLR 2023_

### Official Review · Reviewer_Rmep · 2022-10-24

**Confidence:** 4
**Correctness:** 3
**Technical Novelty And Significance:** 3
**Empirical Novelty And Significance:** 3
**Recommendation:** 5

**Clarity, Quality, Novelty And Reproducibility:**

Some comments to review

_Abstract The claim ”the expressiveness of its representations is severely limited by the anisotropy issue where the hidden representations are distributed into a narrow cone in the vector space” stated in the abstract should be addressed later in the paper

_ Introduction. The claim “the resulting anisotropic representations fail to provide the desired semantic discrimination that is crucial for language understanding tasks, e.g., Semantic Textual Similarity” is arguable. The best existing models for semantic text similarity are currently all based on attention-based models such as transformers.

_ Related Work: “.. the anisotropy issue where the representations are [..] Such degeneration is undesired, especially considering expressiveness capacity
one would expect on modern language models ”. Ok from the first comment. But stating that the degeneration is not desired and it might hurt expressiveness it’s not the same as stating that the degeneration “severely” limits it.

_ Related Work: “However, they suffermost on the discriminative tasks such as code search or clone detection (Lu et al., 2021; Huang et al., 2021; Guo et al., 2022) due to the anisotropy issue (Ethayarajh, 2019)”. First of all, none of the papers analyze the limitations of their performance due to the anisotropy issue. Second, Lu et al., 2021 (aka, CodeXGlue paper) only included Bert-based and GPT-based models for baselines purposes. CodeGPT was not even tested on clone detection nor code search. Third, Guo et al., 2022 presented an encoder- decoder model which surpassed other previous encoder only models.

_ Section 4.3.1 Code Search:  Why the authors choose CodeNet challenge and not CodeXGlue (Lu et al., 2021). More models could have been used to locate where the performance of the ContraGen model.


**Strength And Weaknesses:**

The paper is quite ambitious but is really easy to read, clear and concise.
A vast analysis is included in the Appendix that helps the overall understanding of the approach.
The improvements in terms of Semantic similarity and Code search are significant.

On the weaknesses the two main issues (discussed below) are the justification of the motivation and the results that support it.
Authors argue that the anisotropy is the source of limited performance on generation models but reducing that aspect didn’t bring improvement in terms of PPL for natural language, and a limited improvement in terms of code generation.


**Summary Of The Paper:**

The current research piece introduces a dual contrastive learning objective for training GPT-like (or Causal language ) models (CLM). The main motivation comes from the finding shown in previous work (Ethayarajh, 2019) where the representations in upper layers of CLMs  show a particular clustering (the authors called it anisotropy) that limits the capacity of such representations. Authors also hypothesize that text generation might benefit if such anisotropy is reduced.
The Model, ContraGen, is trained using two contrastive objectives, one token based and one whole sequence based. Two models using each individual strategy are also included for comparison purposes. ContraGen-Tok and ContraGen-Seq
The authors put special emphasis on comparing this approach with SimCTG, a similar approach to ContraGen-Tok with the addition of a temperature in the contrastive loss.
Experimentation is carried out in many aspects, particularly on tasks where semantic discrimation is needed: for natural language, semantic textual similarity (STS) and text generation, and for code, Code Search and Code Completion.
Benchmarks on STS and Code Search show a clear advantage of the current approach against baseline (GPT/CodeGEN), the model furter trained with CLM, and SimCTG.
On the Generation tasks, the current approach cannot surpass CLM in terms of perplexity on natural language but improves over Pass rate on code tasks


**Summary Of The Review:**

Judging the paper from its conclusion “we present ContraGen, an effective contrastive learning framework for language generation  ..to resolve the representation degeneration issue” I think the authors did a good job in solving the latter aspect but I fail to bring enough evidence to prove the former.

---

> ### Author Response · Authors · 2022-11-19
> **Addressing Reviewer Rmep's Comments**
>
> We are glad that the reviewer finds our paper is easy to follow. We also sincerely thank the reviewer for the detailed comments as well as acknowledging the significant improvements we attained on discriminative tasks and the value of the vast analysis we conducted.
>
> **[Comment 1]** On the weaknesses the two main issues are the justification of the motivation and the results that support it. Authors argue that the anisotropy is the source of limited performance on generation models but reducing that aspect didn’t bring improvement in terms of PPL for natural language, and a limited improvement in terms of code generation.
>
> - **[Response]** We have updated our paper to make it clear on our motivation and how our experiments support it. We also summarize it here. We hypothesized that mitigating the anisotropy issue would results in an improved discriminative representation space. Therefore, we highlighted the performance improvements in discriminative tasks and discussed about how anisotropic representations cause poor discriminative representations in Section 4.2.1 and Appendix B.
>
>     At the same time, we showed that generation performance also improves. Note that, ContraGen outperforms CLM on code completion task by 9% in terms of pass@1 accuracy which is a significant improvement as it reflects that more model-generated codes pass a suite of test cases. Improving execution accuracy it is challenging as it requires the model understanding the task, figuring out how to accomplish it, and writing code without any syntax/runtime errors.
>
>     On the natural language, PPL slightly increased due to the regularization effect of contrastive training, but improvements are observed on both MAUVE and the representation quality evaluated with ISim/ESim/Disc.
>
>     Please also refer to our overall rebuttal.
>
> **[Comment 2]** "_Abstract The claim “the expressiveness of its representations is severely limited by the anisotropy issue [..]” stated in the abstract should be addressed later in the paper."
>
> - **[Response]** Thank you for the comment. We discuss what you suggested in the second paragraph of Section 4.2.1 and put the detailed analysis on the anisotropy issue and the poor performance of CLM on discriminative tasks and how ContraGen improves them in Figure 4 in Appendix B. Specifically, we show that the representations yield by CLM are squeezed into a tiny cone where the cosine similarity between both similar and dissimilar sequence pairs are almost at one. In contrast, ContraGen is capable of dispersing the representations apart in the representation space with semantically similar sequences attaining high cosine similarity and dissimilar sequences attaining low cosine similarity.
>
> **[Comment 3]** "_ Introduction. The claim “the resulting anisotropic representations fail to provide the desired semantic discrimination that is crucial for language understanding tasks, [..]” is arguable..."
>
> - **[Response]** The underlying motivation behind the claim is the anisotropy issue that GPT-style models suffer from, which can prevent it from attaining better performance on discriminative tasks. However, as prior works showed that other pretrained transformer models suffer from the anisotropy issue to some extent but some of them are state-of-the-art for STS tasks. So, we agree with the point and therefore we have revised the statement, though we do believe further enhancing the isotropy and discrimination of the existing SOTA models can further boost the performance as well (e.g., as demonstrated in [1]).
>
>     [1] SimCSE: Simple Contrastive Learning of Sentence Embeddings. EMNLP 2021.
>
> **[Comment 4]** "_ Related Work: “.. the anisotropy issue where the representations are [..] Such degeneration is undesired, [..] ”. [..]. But stating that the degeneration is not desired and it might hurt expressiveness it’s not the same as stating that the degeneration “severely” limits it."
>
> - **[Response]** Thank you for the suggestion. We have revised the statements.
>
> **[Comment 5]** "_ Related Work: “However, they suffer most on the discriminative tasks such as code search or clone detection (Lu et al., 2021; Huang et al., 2021; Guo et al., 2022) due to the anisotropy issue (Ethayarajh, 2019)”....."
>
> - **[Response]** This is a misunderstanding. We cited Lu et al., 2021; Huang et al., 2021; Guo et al., 2022 because they studied code search and clone detection tasks and proposed datasets/models. Anisotropy issue is studied in Ethayarajh, 2019.
>
> **[Comment 6]** "_ Section 4.3.1 Code Search: Why the authors choose CodeNet challenge and not CodeXGlue (Lu et al., 2021). More models could have been used to locate where the performance of the ContraGen model."
>
> - **[Response]** We followed [1] to conduct experiments on “zero-shot” code search. CodeXGlue dataset is generally used to evaluated in supervised setting, therefore, we didn’t use it in our study.
>
>     [1] UniXcoder: Unified Cross-Modal Pre-training for Code Representation.

---

### Official Review · Reviewer_9pG3 · 2022-10-24

**Confidence:** 4
**Correctness:** 1
**Technical Novelty And Significance:** 3
**Empirical Novelty And Significance:** 3
**Recommendation:** 3

**Clarity, Quality, Novelty And Reproducibility:**

The paper is novel but the clarity needs further improvement.

More specifically, the model description section is a bit hard to follow. For example, what does h^{j+} mean?

**Strength And Weaknesses:**

Contrastive learning is quite a natural idea for targeting the anisotropy issue. The author has applied the model to various NLP tasks to assess its validity.

This paper could be a good paper about language modelling, but what makes me uncomfortable is that this paper repeatedly says it is about language generation. Nevertheless, except for the code generation experiment that appears at the very end of this paper, this paper does nothing related to language generation. This results in many errors:
1. Semantic textual similarity and code search are not downstream tasks of language generation
2. How the PPL was used in 4.2.2 is actually for evaluating language models
3. More justification is needed for why ISim, ESim and Disc are evaluating generation quality

**Summary Of The Paper:**

To address the problem of the anisotropy issue (i.e., hidden representations of transformer models are squeezed into a tiny cone), this paper proposes to use contrastive learning for language generation (henceforth, ContraGen). The model was evaluated on both text generation and code generation. It was also assessed on tasks like semantic textual similarity and code to code search.

**Summary Of The Review:**

This paper could be a good paper about language modelling, but it is nearly nothing about language generation. Since this paper does something that is inconsistent with what it claims, I cannot recommend an acceptance.

---

> ### Author Response · Authors · 2022-11-19
> **Addressing Reviewer 9pG3's comments**
>
> We thank the reviewer for the comments.
>
> **[Major concern]** "This paper could be a good paper about language modeling, but what makes me uncomfortable is that this paper repeatedly says it is about language generation. Nevertheless, except for the code generation experiment that appears at the very end of this paper, this paper does nothing related to language generation. This results in many errors."
>
> - **[Response]** We want to clearly state the main goal of this work - improving causal (decoder-only) language models' discriminative ability as their rapid expansion in recent times (e.g., GPT-3, Codex) could facilitate language understanding applications (e.g., semantic textual similarity, code search). Since causal language models are typically used in language generation applications, the term “language generation” naturally comes into narratives. Therefore, we request not to misjudge our work. The paper is about as the title says, “effective contrastive learning for causal language model” to show that “it can effectively enhance both uniformity and discrimination of the representations and lead to the desired improvement on various language understanding tasks” as outlined in the abstract. Please also refer to our overall rebuttal above for more on the motivation of the work and how our experiments support it.
>
> **[Comment 1]** "Semantic textual similarity and code search are not downstream tasks of language generation."
>
> - **[Response]** This is a misunderstanding. We didn’t make such statement in the paper. We have updated the paper draft to make it more clear that our main motivation is to make CLM more suitable for discriminative tasks while improving or at least retraining language generation performance.
>
> **[Comment 2]** "How the PPL was used in 4.2.2 is actually for evaluating language models."
>
> - **[Response]** As stated above, ContraGen is designed to improve causal language models such that they generate discriminative language representations which is demonstrated in section 4.2.1. In section 4.2.2, we show that while improving on discriminative tasks, ContraGen does not degenerate in language modeling if not improving. In section 4.3.2, we show the same for the programming language domain using code completion task.
>
> **[Comment 3]** "More justification is needed for why ISim, ESim and Disc are evaluating generation quality."
>
> - **[Response]** Those metrics are indeed proposed for evaluating the representation quality, but it also indicates the generation quality from certain perspectives. Recall that Disc = 1 - ISim / ESim, where ISim refers to intra-similarity defined as the average cosine similarity between a sequence and its constituent tokens, and ESim is defined as the average cosine similarity between the representations of a sequence and each token of another sequence. In our case, we expect
>
>     - lower discrimination score Disc(S) between the ground truth and the generated sequence of the same prompt for better semantic coherence.
>
>     - higher discrimination scores Disc(D) for randomly sampled sequence pairs of the generations in the test set for better isotropic representations and diverse generations.
>
>     We have updated Section 4.2.2 to make it clear.
>
> **[Comment 4]** "More specifically, the model description section is a bit hard to follow. For example, what does h^{j+} mean?"
>
> - **[Response]** We define $h^{j}$ and $h^{j+}$ at the beginning of section 3 (the paragraph before section 3.1), where we call out the main notations. $h^{j}$ and $h^{j+}$ denote the sequence-level representation pairs (referred as positive pairs in contrastive learning) of the sequence $x^{j}$ (a text or code snippet).

---

> > ### Comment · Reviewer_9pG3 · 2022-11-19
> > **Response to author response**
> >
> > Thanks for your response.
> >
> > As I said in my review, I totally agree that your work is about the casual language model, but this does not justify the mis-use of the term "language generation".
> >
> > > Since causal language models are typically used in language generation applications, the term “language generation” naturally comes into narratives.
> >
> > Almost all language models can be used in language generation applications, but we cannot naturally give them all names with "generation" in them, right?
> >
> > There is a bank of works that apply contrastive learning to "real" language generation tasks, and your work is clearly different from what they do. Therefore, concluding something like "we present ContraGEN, an effective contrastive learning framework for language generation" is misleading. (I notice that the phrase "for language generation" was deleted in the updated version but there are still a few similar claims left. For example,  in the general response, the authors said the purpose of this work is to make the language model to "improve language generation performances.")
> >
> > > The paper is about as the title says, “effective contrastive learning for causal language model” to show that “it can effectively enhance both uniformity and discrimination of the representations and lead to the desired improvement on various language understanding tasks” as outlined in the abstract.
> >
> > True. But this is also the place that confused me first when I started reading the paper as you even have a "GEN" in the name of your model (I bet it stands for generation right)?
> >
> > I am pretty satisfied with the authors' response to all my other concerns and most of their responses to other reviewers' concerns also make sense to me. However, at least to me, the use of the term "language generation" is misleading. Considering the term appears in an extremely vital place (i.e., model's name)  and considering that authors seem to insist on this, I personally cannot recommend an acceptance. But if other reviewers are OK with this, I am also fine with an acceptance at last.

---

> > > ### Author Response · Authors · 2022-11-20
> > > **Part 2: Addressing Reviewer 9pG3's Concern**
> > >
> > > Thank you for letting us better understand your concern. We didn’t mean to insist on “language generation” in our rebuttal, but rather made our point why it came into our writing. As you mentioned, we tried to resolve your concern by making several changes. There might be some places we missed to update, which we cannot update now but will do in the camera-ready version. As for the model name, we are considering “ContraCLM” if you or some other reviewers still have concern on our current name “ContraGen”.
> > >
> > > Regarding your concern on our goal to "improve language generation performances" - we intended to evaluate whether language modeling (which we referred as language generation) performance gets deteriorated due to the contrastive learning. Our ultimate goal is to make causal language models more effective in discriminative tasks (due to the availability of large CLMs (e.g., GPT-3, Codex) in recent times) while not compromising if we are not able to improve their performance on downstream language generation tasks. We indeed improved text generation (as shown using MAUVE scores) while attaining significant improvement on code completion tasks (please see our response to other reviewers).
> > >
> > > Lastly, we want to confirm that our intention was not to claim that our work is about language generation applications. One of our baselines, SimCTG used the terminology “text generation” in their work. Since, in our work, we evaluated on text generation in both natural and programming languages, we thought to replace the term “text generation” with “language generation”, to cover both the domains. But we didn’t realize that the use of “language generation” phrase would create confusions and hurt our work so tremendously.

---

### Official Review · Reviewer_Jhu9 · 2022-10-25

**Confidence:** 2
**Correctness:** 4
**Technical Novelty And Significance:** 3
**Empirical Novelty And Significance:** 3
**Recommendation:** 6

**Clarity, Quality, Novelty And Reproducibility:**

I am not very familiar with representation learning or code generation/search research workstreams, thus cannot speak to the novelty of the paper. It seems very clear and of high quality due to the ablations that the authors made to explore some anomalies in results (i.e. ContraGen-Seq doing worse than ContraGen-Tok in STS-B).

**Strength And Weaknesses:**

This work is a step towards making decoder-only models more suited for all tasks. This paper clearly outlines the motivation for improving model representations, defines metrics to do so, and contributes a training regime to improve representation learning.

**Summary Of The Paper:**

One weakness of current language models, especially decoder-only ones, is in lack of expressiveness of representations due to the CLM objective it is retrained on, which poses an issue when used for tasks such as STS or retrieval. The authors suggest augmenting this loss with contrastive loss at both a sequence and token level to mitigate this. Additionally, they suggest three metrics that measures representation quality, and show that their training regime improves upon the CLM only baseline for both code search and STS, while also improving upon source code generation.

**Summary Of The Review:**

To the reader who is somewhat familiar with broader work in PLMs but not specifically in representation learning or code generation/search, this seems like a solid paper with a well-outlined motivation, methodology that makes sense, and results that back up their claims. This work is a step towards making decoder-only models more suited for all tasks (including ones that rely more on representations such as information retrieval), which, if the methodology is novel, would be worth putting in front of researchers.

---

> ### Author Response · Authors · 2022-11-19
> **Addressing Reviewer Jhu9's comments**
>
> We sincerely thank the reviewer for the positive feedback of the paper. If you have any further questions regarding the paper, we hope our general rebuttal or those to other reviewers  can solve them.

---

### Official Review · Reviewer_6wZm · 2022-11-04

**Confidence:** 4
**Correctness:** 3
**Technical Novelty And Significance:** 3
**Empirical Novelty And Significance:** 3
**Recommendation:** 6

**Clarity, Quality, Novelty And Reproducibility:**

The paper is well-written overall, and I understood most of the claims and results. The quality is the paper is very good. The authors do not include their code, so verifying the method's reproducibility is hard.

**Strength And Weaknesses:**

Strength:
- The contrastive loss is very intuitive and builds on prior work.
- The experimental setup and very thorough, and most decisions are justified.
- Experiments are run across different domains (natural language and programming language) and models (GPT2 and CodeGen) to show the generality of the proposed algorithm.
- Ablations studies are strategically done to address concerns regarding surprising results (e.g., CONTRAGEN-SEQ performing worse than CONTRAGEN-Tok)

Weakness:
- Most semantic textual similarity models are Bert-based models. Although this paper focuses on CLM, comparing this algorithm to algorithms that use those models seems more relevant.
- Text generation results lack evaluation metrics: BLEU, BLEURT	BERTScore, Distinct1, Distinct2, rep-2, rep-3, rep-4, diversity, MUAVE,  etc. It is unclear how the proposed metrics compare with the standard generation metrics.
- Natural Language results are missing a baseline algorithm, Unlikelihood training. The Natural language section introduces new metrics, but it is unclear how non-contrastive ideas perform with respect to the new metrics.
- Lack of justification of new metrics for evaluating generation quality.
- The motivation of the paper is not entirely empirically supported.



**Summary Of The Paper:**

This paper proposes a new contrastive learning algorithm to deal with the anisotropy issues of large-scale language model representation. The authors present three variants of their algorithm: CONTRAGEN, CONTRAGEN-Tok, and CONTRAGEN-SEQ, where each variant either includes token loss, sequence loss, or both.

**Summary Of The Review:**

The paper is well-written and straightforward to follow. The ideas in this paper are novel, and the authors empirically justify their claims. However, there are some concerns around the experiment setup, as a list in the weakness part. Because of these concerns, my initial score is borderline accept.

---

> ### Author Response · Authors · 2022-11-19
> **Addressing Reviewer 6wZm's comments**
>
> We sincerely thank the reviewer for pointing out that our paper is straightforward to follow and our ideas are novel and well supported by our experiments and analysis. We also thank the reviewer for the detailed comments, which we try to address below.
>
> **[Comment 1]** "Most semantic textual similarity models are Bert-based models. Although this paper focuses on CLM, comparing this algorithm to algorithms that use those models seems more relevant."
>
> - **[Response]** We discuss CLM and BERT-style models in Section 4.4 (second paragraph) and have more detailed comparisons and discussions reported in Table 6 and Appendix D.1. Although we are aware of the performance gap between GPT-style and BERT-style models, we did not include the results of BERT-style models in the main body of the paper since we are focusing on causal language models. In the final version, if necessary we can add one or two rows for the encoder-based models with the best performance.
>
> **[Comment 2]** "Text generation results lack evaluation metrics: BLEU, BLEURT BERTScore, Distinct1, Distinct2, rep-2, rep-3, rep-4, diversity, MUAVE, etc. It is unclear how the proposed metrics compare with the standard generation metrics."
>
> - **[Response]** Thank you for the suggestion. As mentioned in our response to [Q3] in the overall rebuttal listed above, we have updated Table 2 with the MAUVE and shown ContraGen does outperform CLM.
>
> **[Comment 3]** "Natural Language results are missing a baseline algorithm, Unlikelihood training. The Natural language section introduces new metrics, but it is unclear how non-contrastive ideas perform with respect to the new metrics."
>
> - **[Response]** The SimCTG paper showed that it performs better than unlikelihood training. Since, we included SimCTG as a baseline and attained consistently better performance. We thereby did not further compare ContraGen with unlikelihood training. However, we can do it for the final version if it’s strongly recommended.
>
> **[Comment 4]** "Lack of justification of new metrics for evaluating generation quality."
>
> - **[Response]** Thank you for the comments.  Those metrics are indeed proposed for evaluating the representation quality, but it can also indicate the generation quality from certain perspectives. Recall that Disc = 1 - ISim / ESim, where ISim refers to intra-similarity defined as the average cosine similarity between a sequence and its constituent tokens, and ESim is defined as the average cosine similarity between the representations of a sequence and each token from another sequence. In our case, we expect
>
>     - lower discrimination score Disc(S) between the ground truth and the generated sequence of the same prompt for better semantic coherence.
>
>      - higher discrimination scores Disc(D) between generations of randomly sampled prompts in the test set for more diverse generations and better isotropic representations.
>
>    We updated Section 4.2.2 a bit to make it clear.
>
> **[Comment 5]** "The motivation of the paper is not entirely empirically supported."
>
> - **[Response]** We have updated our paper a bit to make it clear on our motivation and how our experiments support it. We also summarize it here.  We hypothesized that mitigating the anisotropy issue would results in an improved discriminative representation space. Therefore, we highlighted the performance improvements in discriminative tasks and discussed about how anisotropic representations cause poor discriminative representations in Section 4.2.1 and Appendix B.
>
>      At the same time, we showed that generation performance also improves. Note that, ContraGen outperforms CLM on code completion
>      task by 9% in terms of pass@1 accuracy which is a significant improvement as it reflects that more model-generated codes pass a suite
>      of test cases. Improving execution accuracy it is challenging as it requires the model understanding the task, figuring out how to
>      accomplish it, and writing code without any syntax/runtime errors.
>
>      On the natural language, PPL slightly increased due to the regularization effect of contrastive training, but improvements are observed
>      on both MAUVE and representation quality evaluated with ISim / ESim / Disc.
>
>      Please also refer to our overall rebuttal regarding the motivation of our work and how our experiments support it.

---

### Official Review · Reviewer_PcBG · 2022-11-07

**Confidence:** 4
**Correctness:** 3
**Technical Novelty And Significance:** 2
**Empirical Novelty And Significance:** 2
**Recommendation:** 3

**Clarity, Quality, Novelty And Reproducibility:**

This paper is well written and it could be reproduced with some effort.

- What is the x-axis of fig-1? Why not use a bar chart if the x-axis imply a different model?


**Strength And Weaknesses:**

Strength
It introduces contrastive learning objectives into LM and shows it benefits similarity based tasks.

Weakness
The experiment result cast some doubt on the experiment design and modeling assumptions.

- Each new objective shows mixed results in code experiment compared to CLM, and only when combining both objectives does it outperform CLM. It casts doubt on the assumption that reducing anisotropy improves LM quality.
- Natural language experiment (sec 4.2) should include some type of automatic metric (e.g. MAUVE) to evaluate the generation quality.
- Given the performance gap between GPT-style and BERT-style models on similarity-based tasks, shouldn’t the paper include experiments with BERT-style models?


**Summary Of The Paper:**

This paper investigates adding contrastive learning objectives to LM to combat the anisotropy issue, under the assumption that increasing representation discrimination would lead to better LM. It shows a positive impact in similarity based tasks and mixed/unclear impact on LM quality.


**Summary Of The Review:**

The experiment design can be improved and some findings cast doubt on the modeling assumptions.

---

> ### Author Response · Authors · 2022-11-19
> **Addressing Reviewer PcBG's Comments**
>
> We thank the reviewer for the comments. We also thank the reviewer for pointing out that our paper is well-written and the experiments can be reproduced (we will release our code and checkpoints after the final decision of the paper is out).
>
> **[Comment 1]** *"Each new objective shows mixed results in code experiment compared to CLM, and only when combining both objectives does it outperform CLM. It casts doubt on the assumption that reducing anisotropy improves LM quality"*
>
> - **[Response]** We respectfully disagree with this statement. We hypothesized that mitigating the anisotropy issue would result in an improved discriminative representation space. Therefore, we highlighted the performance improvements in downstream discriminative tasks where we conducted evaluations on STS and CodeSearch in the “zero-shot” setting and consistently attain large improvement over CLM no matter using the full model ContraGen or its components, ContraGen-Tok & ContraGen-Seq (see Tables 1&3). Such large improvements clearly indicates the contrastive training objectives help learning a better discriminative representation space and improves LM quality.
>
>     Regarding code completion, we want to point out that it is very challenging to improve execution accuracy, as it requires the model understanding the task, figuring out how to accomplish it, and writing code without any syntax/runtime errors. Therefore, it’s possible that a single component of ContraGen is not enough for improving the execution accuracy. From this perspective, the consistent better performance of ContraGen on HumanEval validates our motivation that ContraGen-Tok and ContraGen-Seq complement each other and hence leads to the better performance of ContraGen.
>
>
> **[Comment 2]** *"Natural language experiment (sec 4.2) should include some type of automatic metric (e.g. MAUVE) to evaluate the generation quality."*
>
> - **[Response]** Thank you for the suggestion. As mentioned in our response to [Q3] in the overall rebuttal listed above, we have updated Table 2 with the MAUVE and shown ContraGen does outperform CLM.
>
>
> **[Comment 3]** *"Given the performance gap between GPT-style and BERT-style models on similarity-based tasks, shouldn’t the paper include experiments with BERT-style models?"*
>
> - **[Response]** We discuss CLM and BERT-style models in Section 4.4 (second paragraph) and have more detailed comparisons and discussions reported in Table 6 and Appendix D.1. Although we are aware of the performance gap between GPT-style and BERT-style models, we did not include BERT-style models in the main body of the paper since we are focusing on causal language models. In the final version, if necessary we can add one or two rows for the encoder-based models with the best performance.

---

### Author Response · Authors · 2022-11-19
**Overall Rebuttal**

We sincerely thank all reviewers for the comments and we have updated our paper accordingly. Below, we address some shared concerns and questions, based on which we also updated our paper a bit to make them clear.

**[Q1]** Motivation of the paper and supporting experiments.

- **[Response]** The main motivation of the paper is to take an initial effort to mitigate the anisotropy issue of the decoder-only causal language models so as to make it suitable for more discriminative tasks and retain/improve language generation performances. We hypothesize that anisotropy issue is one of the factors that limits a representation space to be more discriminative. Therefore, if we could mitigate the anisotropy issue, the CLM would improve in discrimination. We emphasize that there is no standard way to measure how much the anisotropy issue is mitigated. On that account, we measure on CLM's effectiveness on discriminative tasks in the "zero-shot" setting, where we attain 63% and 33% relative improvement on STS and CodeSearch, respectively. Further we conducted comprehensive analyses on how the anisotropy issue hurts the discrimination of representations in Section 4.2.1 and Appendix B.

    While discriminative task performances show the value of our proposal, they do not characterize the learned representation space. Therefore, we propose three metrics to holistically indicate the discriminative nature of the learned representations space. As discussed in Section 4.2.2, the proposed metrics also measure (1) the semantic coherence between the generation and ground truth regarding the same prompt; and (2) diversity of the generation for randomly sampled prompts. Despite slight increase in PPL, the proposed metrics show significant improvements in the representation quality which corroborates the findings of the discriminative task evaluation.

**[Q2]** Given the performance gap between GPT-style and BERT-style models on similarity-based tasks, shouldn’t the paper include experiments with BERT-style models?

- **[Response]** We do discuss CLM and BERT-style models in Section 4.4 (second paragraph) and have more detailed comparisons and discussions reported in Table 6 in Appendix D.1. Although we are aware of the performance gap between GPT-style and BERT-style models, we did not include BERT-style models in the main body of the paper since we are focusing on causal language models. In the final version, if necessary we can add one or two rows for encoder-based models with the best performance.

**[Q3]** Natural language experiment (sec 4.2) should include some type of automatic metric (e.g. MAUVE) to evaluate the generation quality.

- **[Response]** Thank you for the suggestion. We have updated Table 2 with the MAUVE and shown ContraGen does outperform CLM.

**[Q4]** Regarding open-sourcing the code.

- **[Response]** As mentioned in our updated draft, we will release our code and checkpoints after the final decision of the acceptance is out.

---

### Decision · Program_Chairs · 2023-01-20

**Decision:**

Reject

**Justification For Why Not Higher Score:**

- the paper does not refer to retrieval augmented language models which blend generative and contrastive loss already.
- the paper does not show that addressing anisotropy might be fixable by fine tuning.

**Justification For Why Not Lower Score:**

- the paper is clear, the approach is simple and easy to incorporate.


**Metareview: Summary, Strengths And Weaknesses:**

Summary: The similarity between pairs of tokens or sentences induced by the last layer of a language model concentrates in a small range of values (anisotropy). This paper proposes to combine classical language modeling loss with contrastive loss at the sentence and token level. This allows a model covering a wider range of similarity values. Experiments show that this strategy preserves the generative ability of the LM while performing better on similarity and search tasks over English text and computer code.

Strength:
- simple modification to classical neural net training, easy to implement and use.
- clarity, motivations, good description and analysis of the experiments.

Weaknesses:
- many tasks (e.g. classification, retrieval, parsing, tagging) relying on the similarity induced by language model can resort to extra data for fine tuning and it is not clear that anisotropy would be an issue in this case.
- the retrieval literature is ignored although it commonly trains language model with sequence level contrastive loss, often in combination with (pre) training with a likelihood objective (e.g. K. Guu et al REALM 2020, G. Izacard et al Unsupervised Dense Information Retrieval with Contrastive Learning 2021, G. Izacard et al Atlas 2022).


**Summary Of Ac-Reviewer Meeting:**